# Evaluation and Modulation of Gut Microbiome Dysfunction in Chronically Critically Ill Patients: A Prospective Pilot Study

**DOI:** 10.3390/ijms26199778

**Published:** 2025-10-08

**Authors:** Ekaterina Chernevskaya, Ekaterina Sorokina, Petr Polyakov, Kirill Gorshkov, Nadezda Kovaleva, Vladislav Zakharchenko, Natalia Beloborodova

**Affiliations:** Federal Research and Clinical Center of Intensive Care Medicine and Rehabilitology, Petrovka Str., Moscow 107031, Russia; esorokina@fnkcrr.ru (E.S.); p.polyakov@fnkcrr.ru (P.P.); kmgorshkov@fnkcrr.ru (K.G.); nykkova@yandex.ru (N.K.); vzakharchenko@fnkcrr.ru (V.Z.); nvbeloborodova@yandex.ru (N.B.)

**Keywords:** microbiome, microbiota, metabiotics, biomarkers, aromatic microbial metabolites, chronic critical illness

## Abstract

Assessing gut microbiota disturbances for subsequent modulation remains a challenge. This study aims to evaluate the safety and efficacy of a microbiota-oriented strategy in treating patients with chronic critical illness (CCI). This single-center prospective study included chronically critically ill patients, stratified into three groups by severity of microbiota dysfunction. Three different microbiota modulation regimens including metabiotics, enteral, and anaerobic-safe systemic antibiotics were applied subsequently. Forty-three patients with chronic critical illness were included. Mild microbiota dysfunction was present in 49% patients, moderate in 19% and severe in 32%. Monitoring of biomarkers for 14 days confirmed the safety of reducing the pharmacological load in mild to moderate microbiota dysfunction. The microbiota-oriented strategy demonstrated improvements in neurological condition, a decrease in inflammation, and normalization of several hematological and biochemical parameters, without contributing to the activation of opportunistic microorganisms in the intestinal microbiota. The incidence of pneumonia in patients with CCI was reduced significantly during the 28-day observation period. The results of the pilot study suggest the potential benefits of a microbiota-oriented strategy in preventing nosocomial pneumonia in CCI patients.

## 1. Introduction

Severe disturbances of the intestinal microbiota are observed in critically ill patients (CCI), regardless of pathology or age, associated with a dramatic imbalance of “health-promoting” commensal gut bacteria and an increase in potentially pathogenic bacteria [1,2,3]. Dysfunction of the gut microbiota in critical illnesses, similar to damage in other organ systems (such as cardiovascular, pulmonary, and renal systems), is associated with severe adverse clinical outcomes, including mortality, prolonged dependence on life support, and life-threatening nosocomial infections [4,5,6]. However, existing international dysfunction scoring systems (the Multiple Organ Dysfunction Score (MODS), the Sequential Organ Failure Assessment (SOFA) score, and the Acute Physiology and Chronic Health Evaluation II (APACHE II)) designed to assess the severity of a patient’s condition and prognosis in intensive care units (ICUs), do not include microbiota data [7]. Although some studies have attempted to classify gut microbiota as enterotypes specific to ICU patients, such as ICU enterotypes I and II, these classifications have not yet been integrated into routine clinical practice [8].

Previously identified severe disturbances in the taxonomic composition and metabolic processes microbiota in chronically critically ill patients [9] underscore the particular relevance of creating and applying early diagnostic screening methods, as well as modulating the microbiota. Chronic critical illness (CCI) is a complex pathological condition that develops in a particular group of patients after an acute critical illness. The peculiarity of this condition is the long-term dependence of patients on intensive care [10]. According to the definition proposed by Gardner and his colleagues, CCI is characterized by the patient’s stay in the intensive care unit for more than 14 days, while experiencing persistent organ dysfunction, which is confirmed by the SOFA scale on or after the 14th day [11]. Several factors, including multiple courses of antibiotics, enteral nutrition, and systemic inflammation, contribute to the loss of microbial diversity, overgrowth of pathogenic bacteria, and depletion of beneficial microbial species. Based on the high rate of in-hospital mortality associated with nosocomial infection, effective treatment and prevention strategies for infection-associated complications are needed in such patients [12,13].

Current research increasingly confirms the existence of a direct relationship between gut and lung microbiota, known as the gut–lung axis [14,15,16]. This relationship is a complex, bidirectional transport mechanism through which microbial toxins and metabolites produced by both the gut and lung microbiota are exchanged. Given this relationship, disturbances in gut microbiota composition can have a significant impact on the respiratory system’s state, leading to deterioration of respiratory parameters [17]. This is why correction of the gut microbiota is considered a promising therapeutic strategy for preventing or mitigating various respiratory system-related pathological conditions [18,19]. In previous studies using mice, it was demonstrated that the intestinal microbiota functions as a protective mediator during pneumococcal pneumonia. Experimental data showed that mice with disrupted microbiota composition exhibited more pronounced bacterial dissemination, an enhanced inflammatory response, more significant organ damage, and increased mortality compared to control groups of animals with a normal microbiome [20]. These findings underscore the vital role of the intestinal microbiota in protecting the body against pulmonary infections, indicating that maintaining a healthy balance of gut bacteria may contribute to more effective resistance against respiratory tract infections.

In this study, we hypothesize that gut microbiota dysfunction plays a significant role in the development of infectious complications in patients with chronic critical illness. At the same time, a personalized microbiome-oriented treatment strategy (MOST) may improve a patient’s clinical outcomes by targeting gut microbiota dysfunction and then correcting it. This study aimed to evaluate the safety and efficacy of MOST in patients with CCI and to determine whether stratifying patients based on the degree of gut microbiota dysfunction (assessed by the Microbiota Dysfunction Degree) could facilitate more effective and individualized treatment.

## 2. Results

### 2.1. Classification of Microbiota Dysfunctions

We developed an integrated microbiota dysfunction degree that allows for a comprehensive assessment of the gut microbiota’s state and aids in determining the degree of the gut microbiota’s dysfunction in combination with an evaluation of the severity of the patient’s health condition.

The proposed Microbiota Dysfunction Degree (MDD) integrates seven critical parameters (antibiotic exposure, biomarkers, microbial data, and organ failure) into a 0–12-point scale (Table 1).

All points are summed up. The total results are interpreted as the degree of microbiota dysfunction, with 0–4 points indicating mild dysfunction, 5–7 points indicating moderate dysfunction, and 8–12 points indicating severe dysfunction. The therapy mode is selected based on the degree of microbiota dysfunction, as shown in Figure 1.

The fundamental component of all modes is the use of metabiotics to maintain and restore the balance of the microbial community (Figure 1). Each subsequent mode involves an increase in the volume of drugs used, which helps minimize the damaging effect on the intestinal microbiota in mild to moderate microbiota dysfunction, limiting the use of broad-spectrum antimicrobials. The selective regimen involves the use of predominantly enteral antimicrobials that have a minimal effect on anaerobic microorganisms of the microbiome. The systemic regimen consists of a combination of enteral antimicrobials to suppress the metabolic activity of pathogenic microorganisms in the intestines, along with etiotropic antimicrobial therapy (AMT), which utilizes AMTs active against the established infectious agent. The choice of the optimal AMT/combination of AMT is made taking into account the nature of the infection, its localization and severity, as well as the individual characteristics of the patient. The primary objective in developing the optimal treatment regimen was to minimize the pharmacological burden on the body while maintaining high treatment efficacy. Particular attention was paid to preventing relapses of bacterial infections and restoring the normal state of the intestinal microbiota, which was achieved through selection of the minimum number of drugs.

An approach to modulating microbiota dysfunction is based on the degree of microbiota dysfunction (MDD), which ranges from 0 to 12 points. In mild dysfunction (MDD 0–4 points), the focus is on supporting the microbiota using metabiotics (Bactistatin, Actoflor-S). In moderate cases (MDD, 5–8 points), therapy shifts to targeted pathogen suppression, along with restoration of the microbiota. This involves combining metabiotics with selective gut decontamination using non-absorbable agents, typically rifaximin 550 mg twice daily, for broad-spectrum coverage with minimal disruption of anaerobes. Alternatively, antibiotics can be selected according to the spectrum of pathogens, with minimal impact on anaerobes. In severe dysfunction (MDD ≥ 9 points), a systemic regimen combines gut-selective antimicrobials with systemically administered agents selected using a precision medicine framework that includes pathogen identification, resistance profiling, and host factors. Therapeutic decision-making involves real-time monitoring of the microbiota using quantitative molecular assays, enabling the dynamic adjustment of the regimen. The primary goals are threefold: (1) effective pathogen control, (2) minimal disruption of commensal ecosystems, and (3) prevention of dysbiosis-related complications. The Appendix A summarizes the antimicrobial drugs prescribed depending on the selected treatment mode in agreement with the clinical pharmacologist and the attending physician. The MDD scoring system was developed empirically and applied prospectively in this study. The proposed cut-offs (0–4 mild, 5–7 moderate, 8–12 severe) were based on clinical reasoning and biomarker distribution in our cohort but require future validation in independent populations.

### 2.2. Patient Characteristics

This study was a single-center clinical study, performed at the Federal Research and Clinical Center of Intensive Care Medicine and Rehabilitology (Moscow, Russia). Patients admitted to the ICUs were screened for eligibility. After applying the exclusion criteria, 9 nine individuals were excluded, and 43 patients with CCI were included in the final analysis, which was then divided into three groups based on the degree of microbiota dysfunction. A flow chart describing the inclusion of patients in the study is presented in Figure 2.

Baseline demographic and clinical characteristics, including comorbid conditions, as well as ICU-related outcomes, are summarized in Table 2. The patient groups were comparable by in terms of gender and age. Among the enrolled patients, 13 had sustained TBI, 23 were admitted with ischemic (11) or hemorrhagic stroke (9) and subarachnoid hemorrhage (3), and 6 had consequences of anoxic traumatic brain injury (3) and neurosurgical intervention (3).

During the observation period (28 days), no patient died. Mechanical ventilation was required in 7 patients (16%). Vasopressor or inotropic support was administered to 2 patients (4,6%).

### 2.3. Changes in Parameters Compared to Baseline

Notably, all patients with CCI upon admission exhibited significant disturbances in the species composition of their microbiota, as observed in all study groups (Figure 3). No statistically significant changes were found in the species composition of the intestinal microbiota between baseline and day 14 of the study.

“Potentially pathogenic” microorganisms were isolated from more than 50% of patients: *Klebsiella pneumoniae*, *Proteus vulgaris/mirabilis*, *Enterobacter* spp., and *Acinetobacter* spp. More than 60% of CCI patients in all groups had an increased inflammatory coefficient, as determined by the ratio of *Bacteroides* spp. to *Faecalibacterium prausnitzii*. At the same time, more than half of the patients had a decrease in commensal and butyrate-producing bacteria, including *Lactobacillus* spp., *Bifidobacterium* spp., and *Faecalibacterium prausnitzii*. Statistically significant differences were found in the percentage of cases where the number of *Bifidobacterium* spp. was below the reference values (less than 10^9^) between groups 1 and 2 (*p*-value = 0.045). The number of *Staphylococcus aureus* was above the reference values (more than 10^4^) between groups 2 and 3 (*p*-value = 0.042). Was noted to decrease the detection rate above reference values of *Klebsiella pneumonia* and *Clostridium* spp. by 2.6 and 1.7 times, respectively (Appendix A). This change demonstrates positive dynamics in terms of reducing the prevalence of these microorganisms in the samples compared to reference values.

Dysbiosis accompanied pronounced disturbances in the metabolism of aromatic amino acids in patients. Statistically significant changes in the concentrations of aromatic microbial metabolites CCI patients compared with healthy volunteers are presented in Table 3.

The values of BA, PhPA, HBA, and p-HPhAA had the highest diagnostic significance. The concentration of aromatic metabolites and biomarkers was assessed in the patient groups on days 1, 7, and 14 (Appendix A). No statistically significant differences in metabolite concentrations were found between the groups; however, the sum of “sepsis-associated” AMM was higher in group 3 and decreased during treatment (Figure 4).

Dynamic changes in the sum of «sepsis-associated» AMM correlated with the severity of the patients’ condition, and changes in inflammation biomarkers: the level of procalcitonin (PCT) was statistically higher in groups 2 and 3 compared to group 1, IL-6 was several times higher CCI patients of groups 2 and 3, but decreased on days 7 and 14 during the treatment (Appendix A). The CCI patients’ condition remained stable; the clinical and laboratory parameters are presented in (Appendix A).

No statistically significant intergroup differences were identified in the studied parameters; therefore, a combined group analysis of the efficacy and safety of the microbiota-oriented patient management strategy was conducted (Table 4).

The use of a microbiota-oriented strategy contributed to an improvement in the neurological status of CCI patients by day 14 (statistically significant positive dynamics according to the Coma Recovery Scale, Revised, S100 protein concentration), a reduction in inflammation (a decrease in C-reactive protein levels, CRP/albumin ratio), normalization of lymphocyte, NLR and platelet levels, an increase in total protein levels, and an improvement in liver function (ALT, AST).

### 2.4. Clinical Efficacy of a Microbiota-Oriented Strategy

One of the essential indicators in assessing complications in patients is the frequency of pneumonia relapses. We analyzed the frequency of infectious respiratory complications when implementing a microbiota-oriented strategy CCI patients with mild dysfunction compared to CCI patients receiving standard therapy. Patients in the CCI group were comparable in terms of gender, age, clinical, and laboratory parameters (Table 5).

An important result was a statistically significant reduction in the frequency of pneumonia recurrence in patients with CCI in the study group (Figure 5).

Thus, by the 7th day, 12 (31.6%) CCI patients with standard therapy were diagnosed with pneumonia, which required the administration of systemic antibacterial treatment by the attending physicians. In the “MOST” group, pneumonia was diagnosed in only 4 (19%) CCI patients during this study period. On the 14th and 21st days of observation in the control group, this indicator was 19 (50.0%) and 23 (60.5%) cases, while in the main group, against the background of the MOST maintenance regimen, 5 (24%) (*p*-value = 0.04990) and 6 (29%) (*p*-value = 0.019) cases, respectively. At the final stage of the study (day 28), in the control group, antibiotics were prescribed to 27 CCI patients (71.1%) due to the development of pneumonia, while in the study group, this indicator remained targeted. The apparent reduction in pneumonia observed in patients with mild dysfunction receiving the microbiota-oriented strategy should be interpreted with caution, as randomized controlled clinical trials are needed to confirm whether the strategy truly reduces pneumonia risk beyond what can be explained by baseline severity differences.

## 3. Discussion

A significant result of our work was the reduction in the incidence of pneumonia in chronically critically ill patients using a microbiota-oriented strategy. 

The proposed microbiota modulation regimens enable the reduction in antimicrobial therapy in patients where possible. At the same time, with a pronounced microbiota imbalance and the presence of infectious foci, it is impossible to limit oneself to metabiotics alone. It is necessary to suppress “harmful” microorganisms and create favorable conditions for the restoration of one’s own “beneficial” microorganisms using selective antibacterial drugs (analogous to selective decontamination of the digestive tract), aimed at restoring the balance in the microbial community. Selective decontamination has previously been shown to be evidence-based in preventing ventilator-associated pneumonia and is associated with lower hospital morbidity rates [21,22]. The implementation of the antimicrobial therapy control strategy measures also reduced the relative frequency of isolation of ESKAPE group microorganisms in the hospital from 36.5% to 22% (*p* < 0.0001) [23]. The results of studies show that the use of antianaerobic antibiotics, leading to the depletion of the natural tendency of commensal anaerobic reactions in the intestine, negatively affects systemic immunity and is associated with an increased risk of mortality in patients with sepsis [24]. Although anaerobic microorganisms rarely cause sepsis, antibiotics with antianaerobic activity are widely used in the ICU. Moreover, enteric anaerobes may play a protective role in respiratory infections and critical illness [25].

The findings of this study reveal significant disturbances in the gut microbiota composition among CCI patients, which align with previous research on the microbiota composition in this vulnerable patient group [26,27]. Pathogenic Overgrowth observed in over 50% of patients (*Klebsiella pneumoniae*, *Proteus vulgaris/mirabilis*, *Enterobacter* spp., and *Acinetobacter* spp.) is consistent with studies demonstrating the increased prevalence of opportunistic pathogens in critically ill patients [15,28]. This overgrowth may contribute to the development of nosocomial infections and prolonged hospitalization [29,30]. The elevated inflammatory coefficient, determined by the *Bacteroides* spp./*Faecalibacterium prausnitzii* ratio in over 60% of patients, indicates a significant shift towards a pro-inflammatory state. This finding corroborates previous research highlighting the role of this ratio as a biomarker of gut dysbiosis [31,32]. Reduction in beneficial bacteria (*Lactobacillus* spp., *Bifidobacterium* spp., and *Faecalibacterium prausnitzii*) observed in the study is particularly concerning, as these bacteria are known for their immunomodulatory and anti-inflammatory properties [33,34]. The statistically significant differences in *Bifidobacterium* spp. levels between groups 1 and 2 (*p* = 0.045) suggest a progressive deterioration of microbiota composition with disease severity. Metabolic disturbances associated with dysbiosis, particularly in the metabolism of aromatic amino acids, underscore the systemic impact of gut microbiota imbalances. It is important to emphasize that the use of metabiotics is most justified in the intensive care unit. This is due to the fact that metabiotics are structural elements of probiotic microorganisms, their metabolites, and/or signaling molecules with an established chemical formula [35]. 

These components can purposefully correct various physiological processes of the host organism, including metabolic, epigenetic, informational, regulatory, transport, and/or behavioral processes associated with the activity of symbiotic microbiota [36,37]. Metabiotics demonstrate complex therapeutic potential, exhibiting anti-inflammatory, immunomodulatory, and antitumor effects, as well as the ability to regulate blood pressure and reduce oxidative stress [38,39]. During experimental studies, it was found that the use of a metabiotic obtained from the *B. bifidum* strain significantly accelerated the healing process of ulcerative lesions. This effect was achieved through the increased expression of superoxide dismutase (SOD) and glutathione peroxidase (*GPx*) genes, which in turn enhanced the antioxidant potential of the wound surface [40]. “Actoflor-S” enhanced the antagonistic activity of probiotic strains *Lactobacillus acidophilus* D-75 and *Lactobacillus acidophilus* D-76 in vitro, increasing their ability to suppress the growth of pathogenic microorganisms by 2–2.5 times. This effect indicates the inducible nature of the additive in the process of synthesis of bacteriocins—special protein substances that exhibit antibacterial activity [41]. Another study found promising synergistic activity of combinations of metabiotics with various antibiotics against pathogenic *E. coli* and *S. aureus* [42]. Fermented foods may have similar effects. A study showed that safety and improved the Gut Microbiome Wellness Index when using kefir in critically ill patients [43].

It is noteworthy that we have previously demonstrated a decrease in the incidence of pneumonia with the use of an inhalation complex of bacteriophages in chronically critically ill patients, showing comparable effectiveness [44,45]. The use of technologies aimed at supporting the body’s natural defenses by preserving beneficial microflora likely helps suppress the growth and reproduction of pathogenic microorganisms, leading to a decrease in the incidence of infectious complications.

The advantage of this stratification of CCI patients is the determination of the optimal regimen for modulating the patient’s microbiota to correct disturbances in the species and metabolic composition of the intestinal microbiota, the elimination of pathogenic microorganisms, if necessary, as well as its versatility and independence from traditional, time-consuming microbiological methods. Thus, the pilot study yielded results confirming the feasibility of successfully applying one of the antimicrobial therapy technologies without antibiotics, thereby sparing the patient’s microbiota. The results obtained in this study indicate the advisability of a further larger-scale multicenter study of the effects of a comprehensive approach to the treatment of CCI patients using not only metabiotics, but also bacteriophages, enteral antimicrobial drugs that regulate the metabolic activity of the intestinal microbiota, which will improve the rehabilitation prognosis for CCI patients.

Our study had certain limitations. First, the comparison of pneumonia recurrence between the mild dysfunction group and an external, non-randomized control cohort is susceptible to selection bias, since differences in baseline pneumonia risk may have influenced outcomes independently of the intervention. Second, the newly proposed Microbiota Dysfunction Degree (MDD) score is preliminary, and its cut-off thresholds (mild: 0–4; moderate: 5–7; severe: 8–12) were determined empirically without external validation. Third, the sample sizes across groups were unbalanced, particularly the moderate group (*n* = 8), which limits statistical power and requires cautious interpretation of subgroup results. Incomplete assessment of metabolic processes and gut microbiota composition due to the lack of short-chain metabolite measurements and metagenomic sequencing, which could explain the observed effects and complement the interpretation of the results. These limitations should be considered when interpreting the data obtained and planning future studies. Finally, since this was a single-center pilot study not registered in a state clinical trial registry (e.g., ClinicalTrials.gov), the results are preliminary and require confirmation by larger, multicenter, randomized studies.

## 4. Materials and Methods

### 4.1. Patients

Patients were screened for eligibility according to the following criteria:

The inclusion criteria: the patient was over 18 years old, had a chronic critical illness, and informed consent from the patient or closest relatives was obtained for inclusion in the study:

The exclusion criteria: low chance of survival, SAPS II score greater than 65 points, treatment with immunosuppressants or corticosteroids, oncological diseases, signs of systemic severe infection (Sepsis-3 criteria), candidemia.

The study was conducted in accordance with the principles outlined in the Declaration of Helsinki. It was approved by the Ethics Committee of the Federal Research and Clinical Center of Intensive Care Medicine and Rehabilitology (protocol code PP #4/20, dated 22 September 2020). Informed consent was obtained from all patients involved in the study.

The control group (*n* = 38) included patients who received therapy in accordance with the standards of recommendations for the primary and related principles. Diagnosis, prevention, and treatment of pneumonia were carried out in accordance with current clinical guidelines and recommendations. All participants in the group at the time of inclusion in the study did not have clinical, laboratory, and instrumental signs of systemic inflammatory processes, and were indicated for the administration of antimicrobial therapy. Treatment and rehabilitation were carried out by specialists who were not aware of the inclusion of patients in this study.

The recruitment of patients meeting the criteria into the main and control groups was carried out passively, without interfering with the treatment process. Microbiological monitoring and laboratory data were collected only on days 1, 7, and 14, while the incidence of pneumonia recurrence was additionally assessed on days 21 and 28 of the observation period.

### 4.2. Sample Collection

Blood samples were collected in vacuum tubes containing a coagulation activator for the determination of biomarkers and circulating metabolites, specifically sepsis-associated aromatic microbial metabolites (AMM).

Blood samples were collected from patients on the day of admission and on days 7 and 14 at the Federal Research and Clinical Center of Intensive Care Medicine and Rehabilitology (Moscow, Russia). Blood samples from healthy volunteers (*n* = 48) were obtained and described earlier [46]. Blood samples were collected from the peripheral vein for routine laboratory analyses in an anticoagulant-free plastic vacuum tube. Serum was separated by centrifuging the blood at 1500× *g* for 10 min on the day of collection. The serum aliquots were transferred into disposable Eppendorf tubes, frozen, and stored at −35 °C. The total number of serum samples (*n* = 81) included 43 from CCI patients and 48 from healthy volunteers. Stool samples were collected into disposable sterile containers. The containers were transported to the lab and frozen immediately.

### 4.3. Sample Analysis

#### 4.3.1. GC–MS Analysis

A total of 129 blood serum samples were examined using the gas chromatography–mass spectrometry (GC-MS) analyses. For the quantitative analysis of microbial metabolites in samples, gas chromatography coupled with mass spectrometry (GC-MS; Trace GC 1310 gas chromatograph and ISQ LT mass spectrometer, Thermo Electron Corporation, Santa Clara, CA, USA) was employed. Concentrations of the following compounds were determined: benzoic acid (BA), phenylacetic acid (PhAA), phenylpropionic acid (PhPA), phenyllactic acid (PhLA), 4-hydroxybenzoic acid (p-HBA), 4-hydroxyphenylacetic acid (p-HPhAA), 4-hydroxyphenylpropionic acid (p-HPhPA), homovanillic acid (HVA), and 4-hydroxyphenyllactic acid (p-HPhLA). The method achieved a limit of quantification of 0.5 µmol/L for all analyzed compounds, with relative standard deviations ranging from 10% to 30%. Calibration curves showed excellent linearity within the clinically relevant concentration range of 0.5 to 15 µmol/L. Sample preparation and GC-MS analytical conditions were performed according to previously established protocols [47].

#### 4.3.2. Biomarker Analysis

C-reactive protein was determined using a Cobas 6000 biochemical analyzer (Cobas 6000, Roche Diagnostics, Mannheim, Switzerland). The determination of procalcitonin (PCT), interleukin-6 (IL-6), neuron-specific enolase (NSE), protein S100 (S100), and cortisol (CORT) was carried out using the immunochemical method with electrochemiluminescence (Cobas e411, Roche Diagnostics, Basel, Switzerland) according to the attached instructions and set of reagents.

#### 4.3.3. Assessment of Disturbances of Intestinal Microbiota

The gut microbial composition was assessed using *Colonoflor-16* test kits (AlphaLab, St. Petersburg, Russia), which contain reagents for DNA isolation, universal PCR primers targeting total bacterial DNA, and species-specific primers for 23 distinct microbial taxa. Quantitative real-time PCR amplification and detection were performed using the DTprime Real-Time PCR Detection System (DNA-Technology, Moscow, Russia), following the protocol provided by the assay manufacturer (Appendix A). Reference values were obtained for the healthy volunteers without gastrointestinal complaints (age over 14 years) from the kit instructions.

#### 4.3.4. Statistical Analysis

The normality of data distribution was evaluated using the Lilliefors test. As the quantitative variables exhibited significant deviations from a normal distribution, non-parametric statistical methods were employed for subsequent analyses. Continuous variables are presented as medians (Me) with interquartile ranges (IQRs), while categorical variables are reported as absolute frequencies and relative percentages.

For within-group comparisons across repeated measurements (Day 1, Day 7, and Day 14), the Friedman test was conducted, followed by post hoc pairwise comparisons with Bonferroni adjustment to control for Type I error inflation. Categorical variables were compared using the chi-square test or Fisher’s exact test, as appropriate. A two-tailed alpha level of 0.05 was adopted as the threshold for statistical significance.

All analyses were performed using IBM SPSS Statistics (Version 27.0, IBM Corp., Armonk, NY, USA). Visualizations were created using Microsoft Excel (Office 2019, Microsoft Corp., Redmond, WA, USA).

## 5. Conclusions

The absence of a priori sample size calculation results in reduced statistical power. The findings are best regarded as exploratory, guiding the design of future powered studies. In this pilot study, we demonstrated the safety and the potential benefit of a microbiota-oriented strategy in managing gut microbiota disturbances and preventing nosocomial pneumonia in patients with chronic critical illness. By establishing clear criteria for assessing microbiota dysfunction, the approach demonstrated improved clinical outcomes without promoting the development of nosocomial infections. The study revealed an improvement in the neurological status of CCI patients by the 14th day, a decrease in the inflammatory process, normalization of the lymphocyte count, neutrophil-to-lymphocyte ratio (NLR), and platelet levels, an increase in total protein levels, and an improvement in liver function. This highlights the value of individualized microbiota modulation as a therapeutic strategy in chronic critical illness and warrants further investigation in randomized controlled trials.

## Figures and Tables

**Figure 1 ijms-26-09778-f001:**
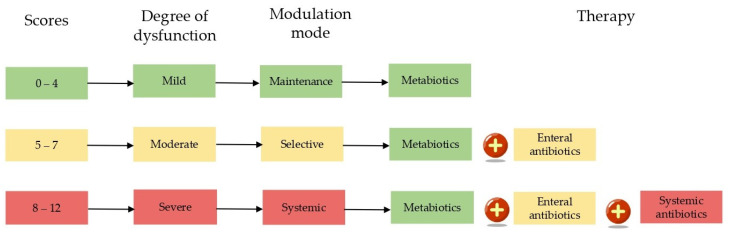
Interpretation of the degree of microbiota dysfunction and selection of modulation mode.

**Figure 2 ijms-26-09778-f002:**
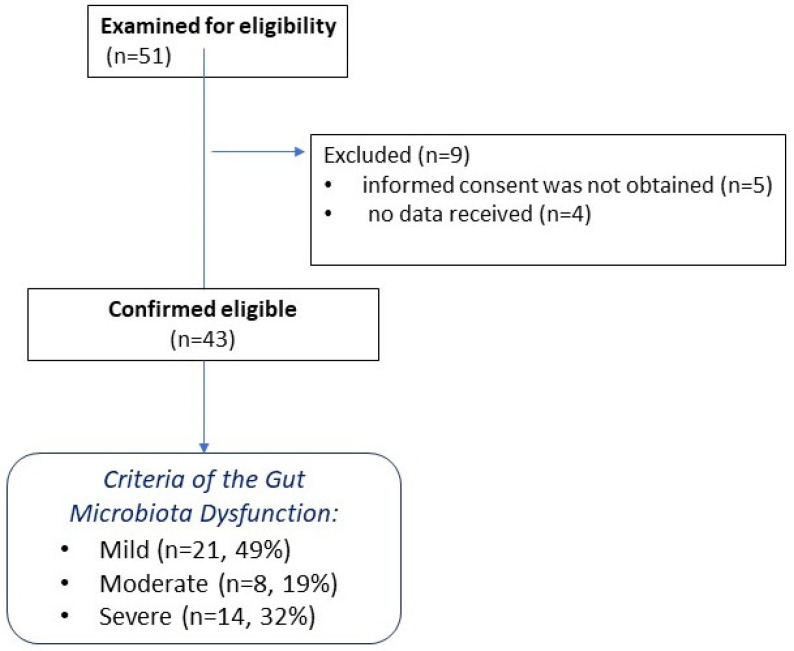
Flowchart of patient selection in the study.

**Figure 3 ijms-26-09778-f003:**
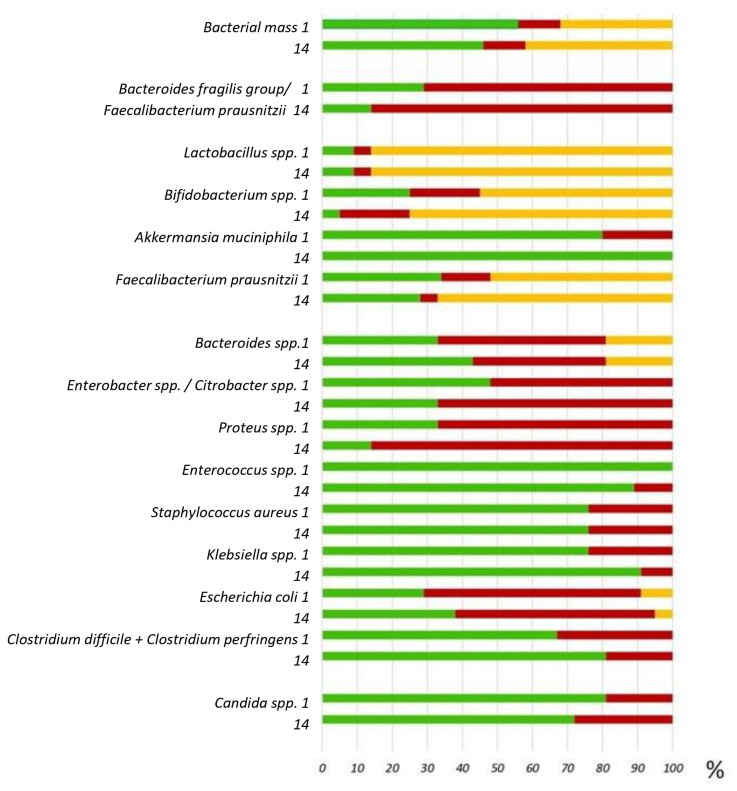
Gut microbiota composition parameters at baseline and on day 14, presented in relation to reference values. The percentage of patients with chronic critical illness whose indicator value is within the reference limits is shown in green. Reference values are exceeded in red, and decreased in orange.

**Figure 4 ijms-26-09778-f004:**
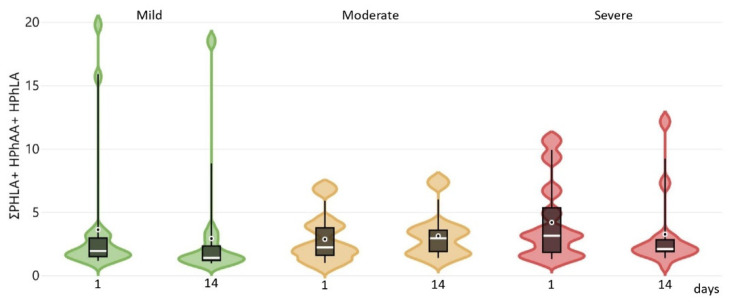
Sum of aromatic microbial metabolites at baseline and on day 14 by three groups by microbiota dysfunctions (mild (*n* = 21)—green color; moderate (*n* = 8)—orange; severe (*n* = 14)—red). The median is shown as vertical lines, the mean value is marked with a dot.

**Figure 5 ijms-26-09778-f005:**
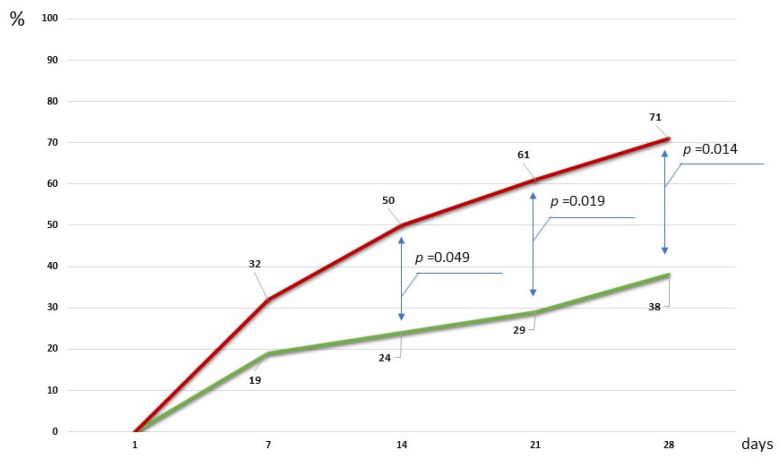
The rate of the incidence of pneumonia recurrence in patients with CCI during the study period; green line—study group; red—control group.

**Table 1 ijms-26-09778-t001:** The calculation of the Microbiota Dysfunction Degree.

	Points		
Parameter	0	1	2
Previous antibiotic therapy	NO	YES	Received upon admission to the ICU
PCT, ng/mL	<0.25	0.25–0.5	>0.5
Presence of a confirmed infectious focus (on CT)	NO	Suspected diagnosis	Confirmed diagnosis
ESKAPE group microorganisms detected at clinically significant titers (>10^4^ CFU/mL)	NO	YES	YES In several locations
Presence of resistance genes	NO	YES	-
Assessment of the colonic microbiota (by PCR/or cultured techniques	Reference values	Deficiency of obligate anaerobes	Deficiency of obligate anaerobes with detection of pathogenic microorganisms at clinically significant titers (>10^4^ CFU/mL)
SOFA score	0–2	>2	-

**Table 2 ijms-26-09778-t002:** Baseline characteristics, comorbidities, and clinical outcomes of ICU patients.

Parameter	Mild (*n* = 21)	Moderate (*n* = 8)	Severe (*n* = 14)	The Pearson Chi-Square or Kruskal–Wallis Tests
Baseline characteristics				
Age, years	58 (40; 65)	53 (38; 58.5)	60 (48; 70)	0.561
Sex (males) n, %	12 (57.1)	4 (50)	8 (57.1)	>0.999
Comorbidities				
CRS-R, points	16 (6; 20)	10 (6; 21)	11 (5; 18)	0.419
Vasoactive drugs, %	1 (4.8)	0 (0)	1 (8.3)	>0.999
Spontaneous breathing, %	16 (76.2)	6 (75)	9 (64.3)	0.805
Spontaneous Breathing with 30% Oxygen, %	1 (4.8%)	0 (0%)	0 (0%)	
BIPAP, %	2 (9.5)	0 (0)	2 (14.3)	
IMV, %	2 (9.5)	2 (25)	3 (21.4%)	
Length of stay, days	56 (43; 76)	54 (38; 71.5)	49.5 (41; 78)	0.770
Length of ICU stay, days	42 (20; 56)	37.5 (25; 59)	29.5 (22; 48)	0.643

Notes: BIPAP, biphasic positive airway pressure (non-invasive ventilation); IMV, invasive mechanical ventilation.

**Table 3 ijms-26-09778-t003:** Aromatic microbial metabolites (AMM) levels in chronic critical illness patients on admission. The values < 0.5 for AMM are for samples with levels below the limit of quantification. Abbreviations: Benzoic acid (BA), phenylpropionic acid (PhPA), phenyllactic acid (PhLA), 4-hydroxybenzoic acid (p-HBA), 4-hydroxyphenylacetic acid (p-HPhAA), 4-hydroxyphenyllactic acid (p-HPhLA), and sum of phenyllactic acid plus 4-hydroxyphenylacetic acid plus 4-hydroxyphenyllactic acid (Σ3AMM).

Parameter	Healthy Volunteers (*n* = 48) Me (Q1; Q3)/N (%) *n* = 48	Patients Me (Q1; Q3)/ *n* = 43	*p*-Value	AUC (95% CI)
BA, µmol/L	0.5 (0.5;0.6)	0.8 (0.7; 1)	<0.001	0.816 (0.706–0.926)
PhPA, µmol/L	>0.5 (>0.5;0.5)	>0.5 (>0.5; >0.5)	<0.001	0.868 (0.789–0.948)
PhLA, µmol/L	>0.5 (>0.5; >0.5)	>0.5 (>0.5; >0.5)	0.080	0.22 (0.17; 0.3)
p-HBA, µmol/L	>0.5 (>0.5; >0.5)	1.7 (>0.5; 3.5)	<0.001	0.919 (0.852–0.985)
p-HPhAA, µmol/L	>0.5 (>0.5; >0.5)	0.9 (>0.5; 2.2)	<0.001	0.887 (0.812–0.962)
p-HPhLA, µmol/L	1.3 (1;1.6)	1 (0.7; 1.2)	0.001	0.712 (0.599–0.825)
Σ3AMM, µmol/L	1.9 (1.5;2.2)	2.1 (1.5; 3.5)	0.037	0.632 (0.507–0.756)

**Table 4 ijms-26-09778-t004:** The summary of the biomarker dynamics, metabolites, clinical and laboratory parameters for 1–14 days (*n* = 43). The values < 0.5 for AMM are for samples with levels below the limit of quantification. The values < 0.02 for PCT and <0.005 for S100 are for samples with levels below the limit of quantification.

Parameter	Day 1 of Admission	Day 7 of Admission	Day 14 of Admission	*p*-Value Friedman Test: Day 1 vs. Day 7 vs. Day 14)	*p*-Value Day 1 vs. Day 7	*p*-Value Day 1 vs. Day 14	*p*-Value Day 7 vs. Day 14
Biomarkers							
PCT, ng/mL	0.13 (0.06; 0.26)	0.07 (0.04; 0.22)	0.08 (0.05; 0.17)	0.151			
CORT, nmol/L	416.3 (268; 826.2)	478.6 (324.6; 552.7)	455.4 (372.4; 576.6)	0.636			
IL-6, pg/mL	31.8 (19.2; 59.9)	21.1 (11.9; 55.2)	23.3 (11.1; 47)	0.492			
S100, mkg/L	0.08 (0.05; 0.12)	0.08 (0.05; 0.19)	0.05 (0.03; 0.1)	0.034	>0.999	0.071	0.085
NSE, ng/mL	12.1 (6.56; 15.2)	12.2 (6.9; 21.8)	15.8 (7.4; 21.6)	0.686			
Assessment Scores							
SOFA	2 (1; 4)	2 (1; 3)	2 (1; 3)	0.059			
CRS-R	12 (6; 20)	12.5 (7; 20)	15.5 (10; 20)	**<0.001**	>0.999	0.114	0.008
CGS	13 (10; 15)	13 (11; 15)	14 (11; 15)	**0.011**	>0.999	0.272	0.826
The blood cells							
WBC, ×10^9^/L	9.1 (6.8; 11,2)	7.5 (6.09; 11,2)	8 (6.3; 11)	0.164			
Lymphocytes, %	14.8 (9.2; 19.9)	18.2 (12.1; 24.2)	22.7 (14.1; 28.4)	**0.002**	0.268	**0.001**	0.185
Neutrophil, %	74.2 (69; 81)	66.8 (59; 77.5)	62 (58.7; 77.3)	0.055			
Platelet, ×10^9^/	295 (225; 399)	307 (255; 338)	335 (281; 381)	**0.010**	0.468	0.341	**0.008**
Neutrophil-to-lymphocyte ratio (NLR)	4.99 (3.52; 8.27)	3.1 (2.24; 5.52)	2.68 (2.06; 5.52)	0.005	0.185	0.004	0.523
	Biochemistry parameters						
Total Protein, g/L	58.8 (55.6; 65)	57.9 (54; 62)	58.65 (55.6; 64.5)	0.025	0.023	0.826	0.341
Albumin, g/L	30.5 (27.8; 33.8)	29 (26.2; 33.4)	31.6 (26.3; 34.1)	0.891			
C-Reactive Protein (CRP), mg/L	48.8 (25.49; 95.7)	41.63 (23.23; 118)	31 (18.5; 52)	0.043	>0.999	0.141	0.061
Urea, μmol/L	4.9 (3.7; 7.1)	4.5 (2.8; 7.7)	4.2 (2.7; 7)	0.888			
Alanine Transaminase (ALT), U/L	34.1 (17.6; 56.5)	28.8 (16.2; 37.5)	18.4 (11.9; 40.3)	0.05			
Aspartate Transaminase (AST), U/L	34.2 (18.3; 49.2)	29.7 (18.4; 43.8)	19 (15.1; 30.8)	0.004	0.149	0.003	0.571
α-Amylase, U/L	68.2 (45.7; 107)	49.8 (40.2; 74.6)	50.7 (38.5; 73.7)	0.137			
C-Reactive Protein/Albumin ratio	1.5 (0.9; 3)	1.5 (0.8; 3.8)	1.1 (0.6; 1.8)	0.025	>0.999	0.247	0.023

WBC—White Blood Cell Count.

**Table 5 ijms-26-09778-t005:** Characteristics of patients.

Parameter	Study Group *n* = 21	Control Group *n* = 38	*p*-Value
Age, years	58 (40; 65)	59 (40.3; 69)	0.511
Sex (female), n%	12 (57.1%)	19 (50%)	0.599
Acute Cerebrovascular Accident, n (%)	10 (48%)	23 (60%)	0.339
Severe Traumatic Brain Injury, %	7 (33%)	9 (24%)	0.425
Anoxia, %	4 (19%)	6 (16%)	0.733
PCT, ng/mL	0.05 (0.03; 0.11)	0.1 (0.06; 0.1)	0.101
C-Reactive Protein (CRP), mg/L	46 (22.5; 83.36)	40.7 (21.6; 105.4)	0.672
Total Protein, g/L	60.5 (57.1; 64)	59.65 (54.37; 66.35)	0.823
Urea, μmol/L	4.8 (2.9; 7.7)	5.3 (3.3; 7.4)	0.674
Creatinine, μmol/L	71.9 (55.6; 87.8)	68.7 (54.6; 83.5)	0.551
White Blood Cell Count (WBC), ×10^9^/L	7.8 (6.4; 9.6)	8.7 (6.4; 10.7)	0.360

## Data Availability

The data presented in this study are available on request from the corresponding author due to privacy and ethical reason of clinical research.

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
