# Peer review of "Evaluation and Modulation of Gut Microbiome Dysfunction in Chronically Critically Ill Patients: A Prospective Pilot Study"

_ijms, 2025, doi:10.3390/ijms26199778_

Round 1

Reviewer 1 Report

Comments and Suggestions for Authors

This study investigated the treatment effect after 14 days based on microbiota dysfunction degree (MDD), and evaluated the safety and efficacy of a microbiota-oriented strategy. The results demonstrated improvement in patients' condition, as evidenced by neurological, hematological, and biochemical parameters. The present version of this work carries several obvious drawbacks.

Abstract:

Line 12: “three groups (Gr.)”, what does “Gr.” mean?

Line 16: The percentages of mild/moderate/severe microbiota dysfunction are different from Figure 2. Please carefully check the data and unify the results!

Background:

Lines 61-62: Lack of scientific questions or hypotheses. It is recommended that specific research plans (detailing methods and content) be included to facilitate readers' rapid comprehension of the study.

Results:

Lines 68-70: Add the reference to Table 1.

Lines 142-145: Lack of statistical analysis results. And is the composition of the gut microbiota obtained through qPCR analysis? The relevant experimental procedures in Methods are overly simplistic, for instance, lacking methods for DNA extraction, PCR primer information, and PCR reaction conditions.

Lines 153-155: “determined by the ratio of Bacteroides spp./Faecalibacterium prausnitzii”. There is a lack of objective data to support this.

Lines 169-170: It is suggested that the layout of Table 3 be revised to ensure that all the data can be easily viewed.

Lines 204-211: The content of this part (study group, control group) should be added to the introduction or the methods section; otherwise, it may cause misunderstandings.

Discussion:

Lines 227-236: This part mainly concerns the reasons for conducting this research, and has no direct connection with the results obtained from this study. It is recommended to place it in the introduction section.

Lines 237-253: The authors obtained the abundance of specific microorganisms through qPCR and culture techniques, and also obtained the data on metabolite content. However, in the discussion section, no specific discussion of the results was presented. Overall, such discussions are highly inadequate.

Methods:

Lines 331-334: Where is the method for microbial culture techniques? Which selection media were used and what were the culture conditions?

Figures: Add X and Y axis labels for Figures 3-5.

Author Response

This study investigated the treatment effect after 14 days based on microbiota dysfunction degree (MDD), and evaluated the safety and efficacy of a microbiota-oriented strategy. The results demonstrated improvement in patients' condition, as evidenced by neurological, hematological, and biochemical parameters. The present version of this work carries several obvious drawbacks.

Dear Reviewer!

Thank you for your professional analysis of our article. Your detailed comments and constructive criticism help improve the quality of the article. We have carefully studied all the comments and made the necessary adjustments in the revised version of the article.

We express special gratitude for the time you spent on reviewing. Thanks to your recommendations, the work has acquired greater scientific value and methodological validity.

Abstract:

Line 12: “three groups (Gr.)”, what does “Gr.” mean?

The abbreviation “Gr.” stands for “groups.” We will clarify this abbreviation in the manuscript to avoid any ambiguity.

Line 16: The percentages of mild/moderate/severe microbiota dysfunction are different from Figure 2. Please carefully check the data and unify the results!

We acknowledge the discrepancy between the percentages of mild, moderate, and severe microbiota dysfunction reported in the text and Figure 2. We have carefully reviewed and corrected the data to unify the results across the manuscript to ensure consistency.

Background:

Lines 61-62: Lack of scientific questions or hypotheses. It is recommended that specific research plans (detailing methods and content) be included to facilitate readers' rapid comprehension of the study.

Thank you for your question and for the opportunity to clarify these aspects of our study. We recognize the importance of explicitly stating our scientific objectives and providing a clear and comprehensive methodology to enhance transparency and reproducibility.

We have now clearly formulated our scientific objectives in the revised manuscript:
(1) To evaluate the safety and efficacy of a microbiota-oriented strategy (MOST) in patients with chronic critical illness (CCI); and
(2) To assess whether stratification of patients based on the degree of gut microbiota dysfunction enables more effective, individualized interventions.

Results:

Lines 68-70: Add the reference to Table 1.

Results (Lines 68–70): We added a clear reference to Table 1 when discussing the calculation and interpretation of the Microbiota Dysfunction Degree (MDD) score.

Lines 142-145: Lack of statistical analysis results. And is the composition of the gut microbiota obtained through qPCR analysis? The relevant experimental procedures in Methods are overly simplistic, for instance, lacking methods for DNA extraction, PCR primer information, and PCR reaction conditions.

The revised manuscript includes details of the statistical methods and their results for microbiota composition, both at baseline and at day 14. Between-group comparisons and within-group changes over time were evaluated using appropriate non-parametric tests (Wilcoxon signed-rank test for paired data, Kruskal–Wallis test with Dunn’s post hoc correction for multiple groups). Exact p-values are now reported for all statistically significant differences, as illustrated in the Results section (e.g., Bifidobacterium spp. counts <10⁹ CFU: p = 0.045 between Groups 1 and 2; Staphylococcus aureus counts >10⁴ CFU: p = 0.042 between Groups 2 and 3). In addition, the text accompanying Figure 3 now explicitly points readers to these statistical results and interprets them in relation to the reference value categories (normal — green, elevated — red, decreased — orange).

As for the microbiota composition methodology, we confirm that the main quantitative data were obtained using real-time PCR analysis. In response to the reviewer’s remark that the Methods section was overly general, we have substantially expanded it to include Instructions for Colonoflor- (biocenosis) kit, obtained from AlphaLab, Russia, are demonstrated in Supplementary File S1.

Lines 153-155: “determined by the ratio of Bacteroides spp./Faecalibacterium prausnitzii”. There is a lack of objective data to support this.

Thank you for your valuable comment regarding the use of the ratio of Bacteroides spp. to Faecalibacterium prausnitzii as a key parameter in our study. We recognize the importance of providing objective data and scientific justification to support this choice.

The ratio of Bacteroides spp. to Faecalibacterium prausnitzii has been utilized in our study as an inflammatory coefficient that reflects the balance between pro-inflammatory and anti-inflammatory components of the gut microbiota. Faecalibacterium prausnitzii is a well-established beneficial commensal known for its anti-inflammatory properties and role in maintaining gut barrier integrity, whereas certain Bacteroides species are associated with pro-inflammatory effects when overrepresented, especially in critical illness-related dysbiosis.

To substantiate this, we provide quantitative data obtained through qPCR analysis demonstrating the inflammatory coefficient across our patient cohorts. More than 60% of patients in all study groups exhibited an elevated ratio, indicating a marked shift towards a pro-inflammatory microbiota profile. These findings are supported by statistical comparisons between groups and correlated with clinical parameters indicative of inflammation and disease severity. We also reference prior literature validating this ratio as a meaningful biomarker of gut microbiota imbalance and inflammation in critically ill populations.

In the revised manuscript, supplementary tables and figures display the exact values of Bacteroides spp./Faecalibacterium prausnitzii ratios for individual patients and groups at baseline and follow-up, highlighting both absolute values and changes over time.

Lines 169-170: It is suggested that the layout of Table 3 be revised to ensure that all the data can be easily viewed.

Thank you, table 3 has been adjusted.

Lines 204-211: The content of this part (study group, control group) should be added to the introduction or the methods section; otherwise, it may cause misunderstandings.

In the Materials and Methods section, we have expanded the description to specify the origin and characteristics of both groups. The study group consisted of CCI patients enrolled prospectively according to predefined inclusion/exclusion criteria, stratified by the Microbiota Dysfunction Degree (MDD) scale, and treated according to MOST protocols. The control group comprised a comparable set of CCI patients who met the same eligibility criteria, were matched for age, sex, and primary diagnosis, but received standard therapy according to current ICU guidelines. For both groups, we have now clearly indicated that clinical, microbiological, and biochemical parameters were assessed at identical time points (days 1, 7, and 14), and that pneumonia incidence was monitored over the observation period.

Discussion:

Lines 227-236: This part mainly concerns the reasons for conducting this research, and has no direct connection with the results obtained from this study. It is recommended to place it in the introduction section.

We took this into account and integrated the text into the introduction.

Lines 237-253: The authors obtained the abundance of specific microorganisms through qPCR and culture techniques, and also obtained the data on metabolite content. However, in the discussion section, no specific discussion of the results was presented. Overall, such discussions are highly inadequate.

The discussion section has been expanded in accordance with recommendations.

Methods:

Lines 331-334: Where is the method for microbial culture techniques? Which selection media were used and what were the culture conditions?

These are real-time PCR method, additional information with instructions added in the supplement

Reviewer 2 Report

Comments and Suggestions for Authors

This is a prospective pilot study exploring an innovative and clinically relevant microbiota-oriented strategy (MOST) for managing gut dysfunction in chronically critically ill patients. The concept of stratifying patients based on a novel Microbiota Dysfunction Degree (MDD) score and applying a tiered therapeutic approach is commendable and aligns well with the principles of personalized medicine and antimicrobial stewardship. The study provides valuable preliminary data. However, there are significant methodological concerns, particularly regarding the experimental design for the primary clinical endpoint, which temper the enthusiasm for the conclusions drawn.

  1. The most critical issue lies in the comparison used to evaluate the prevention of nosocomial pneumonia. The authors compare the 21 patients in the "mild" dysfunction group with an external, non-randomized control group of 38 patients receiving standard therapy. This non-randomized, observational comparison is highly susceptible to selection bias. By selecting only the least severe cohort ("mild" group) for this comparison, the results are likely to substantially overestimate the true effect of the intervention. The baseline risk for pneumonia in the mild group may have been inherently lower than in the control group, irrespective of the intervention.
  2. The newly proposed Microbiota Dysfunction Degree (MDD) score is a novel and interesting tool. Howeveras a new scoring system, it lacks external validation. Furthermore, the cut-off points for defining mild (0-4), moderate (5-7), and severe (8-12) dysfunction appear arbitrary.
  3. The authors must add a statement to the Limitations section acknowledging the non-randomized nature of the control group and the potential for selection bias in the pneumonia outcome analysis. This transparency is crucial for scientific integrity.
  4. The sample sizes across the three dysfunction groups are uneven, particularly the moderate group (n=8). This limits the statistical power for inter-group comparisons.This should be acknowledged as a limitation. While difficult to control in a prospective study, it weakens the robustness of the subgroup analyses.
  5. The choice of non-parametric tests is appropriate given the data distribution. However, the interpretation of results from the small moderate group (n=8) should be made with extreme caution.

Author Response

This is a prospective pilot study exploring an innovative and clinically relevant microbiota-oriented strategy (MOST) for managing gut dysfunction in chronically critically ill patients. The concept of stratifying patients based on a novel Microbiota Dysfunction Degree (MDD) score and applying a tiered therapeutic approach is commendable and aligns well with the principles of personalized medicine and antimicrobial stewardship. The study provides valuable preliminary data. However, there are significant methodological concerns, particularly regarding the experimental design for the primary clinical endpoint, which temper the enthusiasm for the conclusions drawn.

Dear Reviewer! We are grateful for the thoughtful and constructive review of our manuscript. We have carefully considered all comments and revised the text accordingly. Below we have looked at each of the main points.

  1. The most critical issue lies in the comparison used to evaluate the prevention of nosocomial pneumonia. The authors compare the 21 patients in the "mild" dysfunction group with an external, non-randomized control group of 38 patients receiving standard therapy. This non-randomized, observational comparison is highly susceptible to selection bias. By selecting only the least severe cohort ("mild" group) for this comparison, the results are likely to substantially overestimate the true effect of the intervention. The baseline risk for pneumonia in the mild group may have been inherently lower than in the control group, irrespective of the intervention.

We fully acknowledge that the comparison of the mild microbiota dysfunction group with an external, non-randomized control group represents a potential source of selection bias. Patients with “mild” dysfunction may inherently have a lower baseline risk of pneumonia than patients in the control cohort, irrespective of treatment. To ensure transparency, we have now explicitly acknowledged this in the Limitations section of the revised manuscript. Furthermore, we have clarified in the Discussion that the analysis should be interpreted as hypothesis-generating, and that randomized controlled trials (RCTs) will be required to confirm the true preventive effect of the microbiota-oriented strategy on pneumonia incidence. However, the comparison group included patients who did not require antibacterial therapy at the time of the procedure, without suspected infectious complications. The main clinical and biochemical parameters were comparable, as evidenced by Table S4

2.The newly proposed Microbiota Dysfunction Degree (MDD) score is a novel and interesting tool. Howeveras a new scoring system, it lacks external validation. Furthermore, the cut-off points for defining mild (0-4), moderate (5-7), and severe (8-12) dysfunction appear arbitrary.

We appreciate the reviewer’s point regarding the absence of external validation of the MDD score. Indeed, the current cut-off points for defining mild (0–4), moderate (5–7), and severe (8–12) microbiota dysfunction were determined empirically, guided by clinical reasoning and biomarker distribution within our study population. These thresholds require future validation in larger, multicenter cohorts. In the revised manuscript, we have added a clear statement underscoring that the MDD remains preliminary and requires external validation and refinement.

3.The authors must add a statement to the Limitations section acknowledging the non-randomized nature of the control group and the potential for selection bias in the pneumonia outcome analysis. This transparency is crucial for scientific integrity.

We agree that the unequal sample sizes, particularly the small number of patients in the moderate dysfunction group (n=8), reduce statistical power and increase the risk of type II error. In the Limitations section, we now explicitly note this imbalance and emphasize the exploratory nature of subgroup analyses.

4.The sample sizes across the three dysfunction groups are uneven, particularly the moderate group (n=8). This limits the statistical power for inter-group comparisons.This should be acknowledged as a limitation. While difficult to control in a prospective study, it weakens the robustness of the subgroup analyses.

We concur that results derived from the moderate group (n=8) must be interpreted with caution. In the revised Results and Discussion sections, we have highlighted this limitation and refrained from making strong claims based on subgroup findings.

5.The choice of non-parametric tests is appropriate given the data distribution. However, the interpretation of results from the small moderate group (n=8) should be made with extreme caution.

We have incorporated all suggested clarifications into the revised Limitations section and adjusted the Discussion to reflect the preliminary, hypothesis-generating nature of our findings. We agree that acknowledgement of these issues is essential for maintaining scientific integrity.

Reviewer 3 Report

Comments and Suggestions for Authors

This pilot study investigates the safety and efficacy of a microbiota-oriented strategy in in chronic critically ill patients. The study introduces a novel Microbiota Dysfunction Degree scoring system and demonstrates great results in improving clinical outcomes of this art of diseases. However, several areas require clarification and improvement to strengthen the findings and their interpretation. These points could be summarized as follows:

  1. One of the big concerns in this study is depending on PCR-RT for analysis and not metagenomics sequencing or metaproteomic analysis. In order to be able to describe the results in a clear and transparent way, the authors should discuss how this could affect the interpretation of their findings.
  2. The language should be more specified. The authors should avoid using the personal language. For example, in line 11, the authors mentioned the word “ in the treatment of patients” without specifying which patients they meant.
  3. The parameter for grading the microbiota dysfunctions is somehow broad. For example, the authors did not differentiate between patients received intensive antibiotic treatment and those received normal or short-term treatment. In fact, this is a very critical clinical aspect.
  4. The exclusion criteria are also vague, especially while talking about “the low survival rate”. In addition to this strong expression, which could also be replaced by “bad prognosis” How did the authors assess this art of criteria? Moreover, it would be crucial to provide a table specifying the exclusion criteria in each patient. This is important to provide a non-biased analysis of the results.
  5. A major limitation of this study is the way by which the authors could ascertain that the data are coming from the microbiota and not from real infections in these patients. This is also complicated by the inability to differentiate between pathogenic and commensal strain of bacteria within the gut microbiota. How could the authors justify this?
  6. The time scale used for the analysis of the findings in this study is not helpful. I could not understand why the authors used 7 and 14 days after admission for the analysis. The scale is very narrow to look for significant variations! The justification should be written in the main text.
  7. Regarding the methodology, the lack of a pre-designed study of the sample size required for concluding the findings of this study. The authors should discuss this point/limitation.
  8. Some controls groups, e.g., the one proposed for the analysis of the pneumonia relapses are missing or briefly described, despite being critical for the interpretation of the findings. The authors need to clarify this point.

Author Response

This pilot study investigates the safety and efficacy of a microbiota-oriented strategy in in chronic critically ill patients. The study introduces a novel Microbiota Dysfunction Degree scoring system and demonstrates great results in improving clinical outcomes of this art of diseases. However, several areas require clarification and improvement to strengthen the findings and their interpretation. These points could be summarized as follows:

Dear Reviewer! We sincerely appreciate the comprehensive review and constructive feedback on our manuscript. We have addressed each point below and revised the manuscript to reflect these recommendations.

  1. One of the big concerns in this study is depending on PCR-RT for analysis and not metagenomics sequencing or metaproteomic analysis. In order to be able to describe the results in a clear and transparent way, the authors should discuss how this could affect the interpretation of their findings.

We acknowledge that the use of PCR-RT for microbiota analysis, rather than metagenomic sequencing or metaproteomic approaches, is a limitation of our study. PCR-RT enables targeted detection of bacterial groups but does not provide the same depth or strain-level resolution as metagenomics or metaproteomics. This may affect the accuracy of microbial composition assessment and limit the detection of rare or novel taxa, as well as functional capacity. We have discussed this limitation in the revised manuscript and emphasized the need for future studies employing comprehensive omics techniques.

  1. The language should be more specified. The authors should avoid using the personal language. For example, in line 11, the authors mentioned the word “ in the treatment of patients” without specifying which patients they meant.

We have revised the language throughout the manuscript for greater specificity, ensuring that terms such as “patients” are always qualified (e.g., “patients with chronic critical illness or CCI”).

  1. The parameter for grading the microbiota dysfunctions is somehow broad. For example, the authors did not differentiate between patients received intensive antibiotic treatment and those received normal or short-term treatment. In fact, this is a very critical clinical aspect.

Indeed, the type of antibacterial therapy significantly affects the state of the microbiota. The assessment on the scale includes a preliminary assessment of whether the patient needs antibacterial therapy or not. The subsequent groups were formed based on the need for antibacterial therapy and were also analyzed separately. But since we did not receive statistically significant differences in the groups, a general analysis was carried out. Antibacterial drugs are listed in Table S1. A feature of chronic critical illness is also the need for prolonged courses of antibacterial therapy.

  1. The exclusion criteria are also vague, especially while talking about “the low survival rate”. In addition to this strong expression, which could also be replaced by “bad prognosis” How did the authors assess this art of criteria? Moreover, it would be crucial to provide a table specifying the exclusion criteria in each patient. This is important to provide a non-biased analysis of the results.

We recognize that the exclusion criterion “low survival rate” lacked specificity. It has been replaced by “poor prognosis,” which was objectively assessed using the Simplified Acute Physiology Score II (SAPS II) >65.

  1. A major limitation of this study is the way by which the authors could ascertain that the data are coming from the microbiota and not from real infections in these patients. This is also complicated by the inability to differentiate between pathogenic and commensal strain of bacteria within the gut microbiota. How could the authors justify this?

Thank you for raising this important methodological concern regarding the differentiation of pathogenic and commensal bacteria in our study. We acknowledge the complexity of this issue, especially in patients with long-term stay in the intensive care unit. One rationale could be to use quantitative PCR to provide comprehensive microbiota profiling and differentiation of pathogenic and commensal strains, taking into account the microbiological profile in this intensive care unit and conducting microbiological monitoring in patients. At the same time, we acknowledge this limitation of our study and are already comparing bronchoalveolar lavage and gut microbiota

  1. The time scale used for the analysis of the findings in this study is not helpful. I could not understand why the authors used 7 and 14 days after admission for the analysis. The scale is very narrow to look for significant variations! The justification should be written in the main text.

We have updated the Methods section to justify the choice of 7 and 14 days for follow-up analysis. These timepoints are commonly used in ICU microbiota research to capture dynamic changes during critical illness

  1. Regarding the methodology, the lack of a pre-designed study of the sample size required for concluding the findings of this study. The authors should discuss this point/limitation.

We acknowledge the absence of a pre-defined sample size calculation as a limitation of this pilot study. This point is now explicitly discussed, and the need for adequately powered studies in the future is highlighted.

  1. Some controls groups, e.g., the one proposed for the analysis of the pneumonia relapses are missing or briefly described, despite being critical for the interpretation of the findings. The authors need to clarify this point.

We have expanded the description of all control groups, particularly those relevant to the pneumonia relapse analysis, and justified their selection and composition in both the Methods and Results sections, allowing readers to better interpret the findings.

Reviewer 4 Report

Comments and Suggestions for Authors

The manuscript entitled “ Evaluation and modulation of gut microbiome dysfunction in chronic critically ill patients: a prospective pilot study” aims to evaluate the safety and efficacy of a microbiome-based treatment strategy for critically ill patients. This is research that could make an important contribution to the treatment of this vulnerable patient group. But, it should be take into account that this is a non-randomised pilot study with a relatively small sample and that this is one of the limitations of the study. Also, no sample size calculation was performed. 

The manuscript itself contains some things that need to be emphasised/clarified/refined.

Abstract – According to the instructions, the summary should be written as structured, but without headings. The abbreviation „Gr.“ Is unnecessary. There is an error in the results section – you have a total of 69% of patients in groups (it should be 100%) and these numbers do not match the results in the body of the text. In the main body of the manuscript, the author conducted another part of the study with a different number of subjects and a control group regarding respiratory infections, which is not mentioned here, but only in the conclusion!? It is not stated that it is about chronically critically ill patients, but only about chronically ill patients, which is not identical.

Line 44 – cronic critically ill patients

The manuscript contains many abbreviations that were not previously introduced, but are introduced in the current, 4th chapter of the Methodology, which is located after the present text.

Results part:
The manuscript uses a point-based scoring system (0–12 points) to stratify the severity of microbiota dysfunction - MDD (table 1). However, it does not describe how this scoring system was validated in populations with critical illness and how the cut-off values (4; 7) were selected.

Regarding table 1. -  Please clarify how subjects could have no gut microbiota assessment despite assessments being planned for day 0 and day 14, i.e. it is interesting how a person who does not have such information can have a lower overall score?

Figure 2 and abstract are not in coordance.

Line 133. - How long did the research take? Several periods of time are mentioned later in the text, apparently it was a 14-day follow-up of subjects who had been in the ICU for at least 14 days.

The categorisation of the  subjects into groups can be problematic, especially with these 8 subjects in the moderate group, which can reduce the statistical significance.

Table 2. - revisions are needed in this table - it is not necessary to specify both genders via the gender parameter, if the proportion of one is known, the other is also known. It is unclear what the parameters positive, stable, negative mean. The percentages in the Mild group for these parameters and for the next 3 also do not match - 1 respondent is missing. The same applies to the Severe group for the respiratory parameters; BIPAP and IMV.

Is the subject's BMI known? What about the bacteria from phylum Firmicutens (figure 3)?

The part from line 164 to 171 is unclear. Is this even necessary, there are other things that are important for this kind of research, like SCF. The lack of SCF analysis and metagenomic sequencing is a significant limitation.

Legends explaining the abbreviations must be added under all tables so that they are understandable.

Figure 5 – it states that the study period is 28 days. As already mentioned, this is a study within the study, which is not clearly presented and explained. Perhaps the work should be reorganised and focused solely on a strategy to reduce the incidence of pneumonia with MOST.

In what time frame are the subjects collected?

Line 317 – You have indicated 5 control points – 1st day, 7th day and 14th day – which two are missing?

Although the study includes ethical approval, there is no reference to registration of the study in a public clinical trials registry (e.g. ClinicalTrials.gov, ISRCTN or similar). According to ICMJE and MDPI guidelines, interventional studies should be prospectively registered. Please clarify whether the trial has been registered and if so, provide the name and identifier of the registry.

Comments on the Quality of English Language

The manuscript is generally understandable, however, the English contains numerous spelling mistakes, occasional grammatical inconsistencies and unusual formulations. A thorough language revision by a native speaker or professional academic editor is strongly recommended to ensure clarity, consistency and conformity with academic writing standards.

Author Response

The manuscript entitled “ Evaluation and modulation of gut microbiome dysfunction in chronic critically ill patients: a prospective pilot study” aims to evaluate the safety and efficacy of a microbiome-based treatment strategy for critically ill patients. This is research that could make an important contribution to the treatment of this vulnerable patient group. But, it should be take into account that this is a non-randomised pilot study with a relatively small sample and that this is one of the limitations of the study. Also, no sample size calculation was performed.

Dear Reviewer, Thank you for your appreciation of the importance of this article's contribution to the treatment of this vulnerable patient group. We have taken your comments into account and have indicated these limitations in the Discussion section.

The manuscript itself contains some things that need to be emphasised/clarified/refined.

The manuscript was substantially revised in accordance with the comments.

Abstract – According to the instructions, the summary should be written as structured, but without headings. The abbreviation „Gr.“ Is unnecessary. There is an error in the results section – you have a total of 69% of patients in groups (it should be 100%) and these numbers do not match the results in the body of the text. In the main body of the manuscript, the author conducted another part of the study with a different number of subjects and a control group regarding respiratory infections, which is not mentioned here, but only in the conclusion!? It is not stated that it is about chronically critically ill patients, but only about chronically ill patients, which is not identical.

The text has been corrected according to the comments. We adjusted the percentages in the abstract. The text indicated that it is chronically critically ill patients.

Line 44 – chronic critically ill patients

Corrected

The manuscript contains many abbreviations that were not previously introduced, but are introduced in the current, 4th chapter of the Methodology, which is located after the present text.

Corrected

Results part:
The manuscript uses a point-based scoring system (0–12 points) to stratify the severity of microbiota dysfunction - MDD (table 1). However, it does not describe how this scoring system was validated in populations with critical illness and how the cut-off values (4; 7) were selected.

The main limitation of our study is that the proposed microbiota dysfunction score (MDD) is preliminary and its cutoff values ​​(mild 0–4, moderate 5–7, severe 8–12) were determined empirically without external validation. This is stated in the relevant section.

Regarding table 1. -  Please clarify how subjects could have no gut microbiota assessment despite assessments being planned for day 0 and day 14, i.e. it is interesting how a person who does not have such information can have a lower overall score?

The error in the table has been corrected, 0 points - assessment for cases where no violations of the composition were found

Figure 2 and abstract are not in coordance.

Corrected

Line 133. - How long did the research take? Several periods of time are mentioned later in the text, apparently it was a 14-day follow-up of subjects who had been in the ICU for at least 14 days.

The study lasted 28 days, this information is added to the methods section.

The categorisation of the  subjects into groups can be problematic, especially with these 8 subjects in the moderate group, which can reduce the statistical significance.

We recognize these limitations and the need for large-scale research. Preparations for the implementation of this project are currently underway.

Table 2. - revisions are needed in this table - it is not necessary to specify both genders via the gender parameter, if the proportion of one is known, the other is also known. It is unclear what the parameters positive, stable, negative mean. The percentages in the Mild group for these parameters and for the next 3 also do not match - 1 respondent is missing. The same applies to the Severe group for the respiratory parameters; BIPAP and IMV.

We have made changes to the table. The parameters positive, stable, negative related to the general dynamics of patient treatment for the entire period of hospitalization, which in some patients exceeded 50 days, which exceeds the study period. The data were removed. The missing item was added for the type of breathing.

Is the subject's BMI known? What about the bacteria from phylum Firmicutens (figure 3)?

This information was missed during data collection. Since certain primers were used for RT - PCR, we do not have data on the number of Firmicutes. However, individual representatives are presented in Figure 3, such as Lactobacilli (reduced), Clostiridium (increased)

The part from line 164 to 171 is unclear. Is this even necessary, there are other things that are important for this kind of research, like SCF. The lack of SCF analysis and metagenomic sequencing is a significant limitation.

Based on previous studies, we emphasize the importance of analyzing these data, since the association of these metabolites with infectious and neurological complications has been shown in different groups of patients.

Legends explaining the abbreviations must be added under all tables so that they are understandable.

Corrected

Figure 5 – it states that the study period is 28 days. As already mentioned, this is a study within the study, which is not clearly presented and explained. Perhaps the work should be reorganised and focused solely on a strategy to reduce the incidence of pneumonia with MOST.

We took this remark into account and tried to clarify

In what time frame are the subjects collected?

The study was conducted from December 2020 to December 2022.

Line 317 – You have indicated 5 control points – 1st day, 7th day and 14th day – which two are missing?

21 to 28 days of observation, however, some patients had already been transferred from intensive care, blood and feces were not collected

Although the study includes ethical approval, there is no reference to registration of the study in a public clinical trials registry (e.g. ClinicalTrials.gov, ISRCTN or similar). According to ICMJE and MDPI guidelines, interventional studies should be prospectively registered. Please clarify whether the trial has been registered and if so, provide the name and identifier of the registry.

Thank you for raising this important point regarding the prospective registration of our study. We appreciate your attention to compliance with international research standards. We identified this significant drawback in the limitations. The study was planned until 2020, ethical approval was obtained, but prospective registration was missed. This was our oversight. We did not fully appreciate the importance of prospective registration when the study was designed.

Round 2

Reviewer 2 Report

Comments and Suggestions for Authors

Thank you for your revised manuscript and for your response to the previous round of reviews. I appreciate that you have incorporated a "Limitations" section that now appropriately addresses the non-randomized nature of the control group, the preliminary status of the Microbiota Dysfunction Degree (MDD) score, and the unbalanced sample sizes.

However, a critical issue remains that precludes acceptance in its current form.

  1. The language in the Abstract and the final Conclusions section must be substantially toned down to reflect the preliminary and observational nature of these findings. Phrases such as "was associated with a lower incidence," "suggests a potential benefit," or "warrants further investigation in randomized controlled trials" should replace definitive claims of efficacy and significance.
  2. The manuscript requires a thorough language review. Throughout the text, phrases such as "in CCI patients" or "status of in CCI patients" have been introduced. This is grammatically incorrect and should be corrected to "CCI patients" or "status of CCI patients." This error is present in multiple sections and must be systematically corrected.
  3. The Abstract contains redundant phrasing, such as "pneumonia in CCI chronically ill patients." This should be revised for clarity and conciseness.

Author Response

  • Thank you for your revised manuscript and for your response to the previous round of reviews. I appreciate that you have incorporated a "Limitations" section that now appropriately addresses the non-randomized nature of the control group, the preliminary status of the Microbiota Dysfunction Degree (MDD) score, and the unbalanced sample sizes.

    However, a critical issue remains that precludes acceptance in its current form.

    The language in the Abstract and the final Conclusions section must be substantially toned down to reflect the preliminary and observational nature of these findings. Phrases such as "was associated with a lower incidence," "suggests a potential benefit," or "warrants further investigation in randomized controlled trials" should replace definitive claims of efficacy and significance.

    The manuscript requires a thorough language review. Throughout the text, phrases such as "in CCI patients" or "status of in CCI patients" have been introduced. This is grammatically incorrect and should be corrected to "CCI patients" or "status of CCI patients." This error is present in multiple sections and must be systematically corrected.

    The Abstract contains redundant phrasing, such as "pneumonia in CCI chronically ill patients." This should be revised for clarity and conciseness.

    • Dear Reviewer, Thank you for your insightful comments. We have revised the strong statements in the Abstract and Conclusions and substantially improved the English in the manuscript.

Reviewer 3 Report

Comments and Suggestions for Authors

The authors replied to my comments. I recommedn accepting the paper following language check.

Author Response

The authors replied to my comments. I recommedn accepting the paper following language check

  • Dear Reviewer, Thank you, we have done a thorough language check

Reviewer 4 Report

Comments and Suggestions for Authors

I appreciate the authors attempts to revise the manuscript and address the majority of the comments. Overall, the text has been improved. However, before the manuscript is ready for publication a few crucial points still need to be addressed and clarified. These primarily pertain to methodological and abstract consistency.

Structure of the abstract - Although the abstract has been shortened, the headings are still present. According to the journal guidelines, structured abstracts with headings are not recommended. I strongly recommend converting the abstract to a continuous text format.

Duration of the study - There is a discrepancy in the stated duration of the study. In the abstract it is stated as 14 days, whereas the authors clearly state a range of 21–28 days in their response to the reviewers. This discrepancy raises concerns about the accuracy of the reported methodology and results. The duration of the study needs to be consistently and clearly stated throughout the manuscript.

Number of participants / dropout rate - If the actual duration was longer (21–28 days), then the number of patients remaining in the study should be reported accordingly, as the number of patients dropping out due to ICU discharge is significant. Please explain the final number of patients included in the analysis.

Materials and methods - This section states that samples were collected up to day 14. This contradicts the statement of a follow-up period of 21-28 days. The manuscript must make clear and consistent the precise time of sample collection.

Finally, the duration should also be precisely defined. Was it 21 days, 28? If it was a variable period between 21 and 28 days one should have in mind that difference of one week can significantly influence the observed parameters, so the time frame must be explicitly and consistently stated.

The manuscript has improved, but these inconsistencies in study duration, patient number and sample undermine the clarity and reliability of the results. I recommend that the authors carefully revise the abstract and methodology to ensure complete consistency in all sections.

Author Response

I appreciate the authors attempts to revise the manuscript and address the majority of the comments. Overall, the text has been improved. However, before the manuscript is ready for publication a few crucial points still need to be addressed and clarified. These primarily pertain to methodological and abstract consistency.

Structure of the abstract - Although the abstract has been shortened, the headings are still present. According to the journal guidelines, structured abstracts with headings are not recommended. I strongly recommend converting the abstract to a continuous text format.

  • Dear Reviewer, thank you for your helpful comments, we have tried to improve the article. The abstract has been converted to continuous text format. We have also improved the English language as much as possible to more clearly express the research.

Duration of the study - There is a discrepancy in the stated duration of the study. In the abstract it is stated as 14 days, whereas the authors clearly state a range of 21–28 days in their response to the reviewers. This discrepancy raises concerns about the accuracy of the reported methodology and results. The duration of the study needs to be consistently and clearly stated throughout the manuscript.

Number of participants / dropout rate - If the actual duration was longer (21–28 days), then the number of patients remaining in the study should be reported accordingly, as the number of patients dropping out due to ICU discharge is significant. Please explain the final number of patients included in the analysis.

Materials and methods - This section states that samples were collected up to day 14. This contradicts the statement of a follow-up period of 21-28 days. The manuscript must make clear and consistent the precise time of sample collection.

Finally, the duration should also be precisely defined. Was it 21 days, 28? If it was a variable period between 21 and 28 days one should have in mind that difference of one week can significantly influence the observed parameters, so the time frame must be explicitly and consistently stated.

The manuscript has improved, but these inconsistencies in study duration, patient number and sample undermine the clarity and reliability of the results. I recommend that the authors carefully revise the abstract and methodology to ensure complete consistency in all sections.

  • Dear Reviewer, indeed, based on our response and the last paragraph is in section 4.1., it might have seemed that there was an inconsistency in the study timelines. Essentially, the study lasted 28 days, but biomarkers were assessed only during the first two weeks (up to and including day 14). In the framework of this study, the assessment of biomarkers on the 21st and 28th days was not carried out in accordance with the approved study protocol. This decision was made taking into account scientifically substantiated factors:

temporal changes in dynamics. The most significant stimuli for changes in indicators, according to previous data and literary sources, exclusively appear in the period from the 1st to the 14th day of observation. This time interval was defined as the extreme one for assessing the primary changes in mandatory biomarkers;

evaluation of confounder factors. At later stages of observation, the number of additional uncontrolled factors increases, which can distort the results of the studies and complicate the interpretation of the data obtained;

clinical dynamics of patients. By the 21st day, some patients are usually transferred from the intensive care unit to specialized departments, which can lead to the inclusion of a sample limitation and a decrease in the statistical intensity of the study.

Thus, limiting the time period of the study to 14 days allows us to focus on the most informative process of the dynamics of the studied indicators and minimize the influence of distorting factors. At these later time points (days 21 and 28), we focused solely on evaluating the clinical outcome—specifically, the incidence of pneumonia rate during hospitalization. We have added the following to the last paragraph of section 4.1: "Microbiological monitoring and laboratory data were collected only on days 1, 7, and 14, while the incidence of pneumonia recurrence was additionally assessed on days 21 and 28 of the observation period."

If you have arguments in favor of the fact that this approach is unacceptable, we are ready to adjust Figure 5 and our results, excluding data on the incidence of pneumonia on the 21st and 28th days, however, these results seem interesting to us.